# Is There a High Risk for GI Bleeding Complications in Patients Undergoing Abdominal Surgery?

**DOI:** 10.3390/jcm12041374

**Published:** 2023-02-09

**Authors:** Dörte Wichmann, Olena Orlova, Alfred Königsrainer, Markus Quante

**Affiliations:** 1Department of General, Visceral and Transplant Surgery, University Hospital Tübingen, Hoppe-Seyler-Str. 3, 72076 Tübingen, Germany; 2Medical Clinic, Mühlacker Hospital, Hermann-Hesse-Strasse 34, 75417 Mühlacker, Germany

**Keywords:** postoperative gastrointestinal bleeding, bleeding after GI surgery, endoscopic complication management

## Abstract

Introduction: Gastrointestinal bleeding (GIB) can cause life-threatening situations. Here, endoscopy is the first-line diagnostic and therapeutic mode in patients with GIB among further therapeutic approaches such as embolization or medical treatment. Although GIB is considered the most common indication for emergency endoscopy in clinical practice, data on GIB in abdominal surgical patients are still scarce. Patients and methods: For the present study, all emergency endoscopies performed on hospitalized abdominal surgical patients over a 2-year period (1 July 2017–30 June2019) were retrospectively analyzed. Primary endpoint was 30-day mortality. Secondary endpoints were length of hospital stay, cause of bleeding, and therapeutic success of endoscopic intervention. Results: During the study period, bleeding events with an indication for emergency endoscopy occurred in 2.0% (129/6455) of all surgical inhouse patients, of whom 83.7% (*n* = 108) underwent a surgical procedure. In relation to the total number of respective surgical procedures during the study period, the bleeding incidence was 8.9% after hepatobiliary surgery, 7.7% after resections in the upper gastrointestinal tract, and 1.1% after colonic resections. Signs of active or past bleeding in the anastomosis area were detected in ten patients (6.9%). The overall 30-day mortality was 7.75%. Conclusions: The incidence of relevant gastrointestinal bleeding events in visceral surgical inpatients was overall rare. However, our data call for critical peri-operative vigilance for bleeding events and underscore the importance of interdisciplinary emergency algorithms.

## 1. Introduction

Gastrointestinal bleeding (GIB) is considered the most common indication for emergency endoscopy in clinical practice. According to the current literature, the incidence of GIB is 47/100,000 in the upper gastrointestinal tract (UGIT) or 33/100,000 in the lower gastrointestinal tract (LGIT) [1]. Depending on the cause of bleeding, localization and severity, GIB can lead to life-threatening situations and is associated with a mortality of 2–10% (UGIT) [2] and 2.4–3.9% (LGIT) [3], respectively. However, it must be considered that the respective patient populations studied mostly consist of gastroenterological patients. In contrast, the incidence, cause and therapy of GIB explicitly in surgical patients are not well studied in the current literature. Here, single studies are demonstrating an increased procedure-specific bleeding risk after surgery [4]. However, surgical patients are often older compared to the general population and commonly in a reduced general condition due to previous treatments (radio; chemotherapies) and surgical interventions, possibly with anastomoses in the GIT.

For the emergency management of GIB, endoscopic diagnosis and immediate therapy, if possible in one session, are the current gold standard. While numerous studies have investigated the incidence and causes of gastrointestinal bleeding events in gastroenterological patients, data on GIB in surgical patients are scarce. Therefore, in the present study, we retrospectively analyzed our complete surgical patient population over a period of two years in regard to the incidence of GIB.

## 2. Materials and Methods

For the present study, all emergency endoscopies of inpatients in the Department of General, Visceral and Transplant Surgery at the University Hospital of Tübingen over a period of 2 years (1 July 2017–30 June 2019) were retrospectively analyzed. The local ethics committee approved the study (922/2018BO2), and the project was registered as a clinical trial (NCT04523753). 

Inclusion criteria were inpatient care by the surgical department, indication for emergency endoscopy due to gastrointestinal bleeding and patient age > 18 years. The inclusion and exclusion criteria are listed in Table 1. For the definition of gastrointestinal bleeding, the parameters according to the DGCS “S2k-Guideline gastrointestinal bleeding” [5] were used as the basis for the analysis.

Other aspects that were recorded and evaluated for the analysis are the clinical course, previous diseases, current medications, type of surgical intervention and the endoscopic findings as well as the success of the endoscopic therapy. The type of surgical care was described by dividing into four surgical areas (Table 2). The primary endpoint was 30-day mortality. Secondary endpoints were intensive care unit treatment duration, cause of bleeding, and therapeutic success of endoscopic intervention. 

This study is a descriptive analysis. The statistical analysis of the data collected, as well as the graphs and tables presented in the paper, were created using Microsoft’s Excel spreadsheet software. The data are presented as absolute numbers or as means with standard deviation.

## 3. Results

During the study period, bleeding events with indication for emergency endoscopy occurred in 2.0% (129/6455) of all surgical inhouse patients. Of these 129 patients, a total of 83.7% (*n* = 108) underwent surgery, while 21 patients (16.0%) underwent emergency endoscopy on the surgical ward without documented surgical procedures during the same inpatient stay. Patient’s characteristics are shown in Table 3. 

The patients without surgical procedure were excluded from further analyses. Of the analyzed 108 patients undergoing surgery and emergency endoscopy for suspected GIB events, *n* = 94 (87.01%) were examined after surgery. A total of 14 (12.96%) patients underwent endoscopy prior to surgery, where endoscopy was leading the indication for surgery in more than 50% (8/14) of the patients. 

However, for the vast majority of our inhouse patients (*n* = 94/108; 87.01%), emergency endoscopy due to suspected GIB took place after surgery. In detail, fifty-one patients (47.22%) underwent surgery on the LGIT, 26 patients (24.70%) underwent surgery on the Hepato-pancreatico-biliary (HPB) system, 23 patients (21.29%) underwent surgery on the UGIT, and eight patients underwent surgery that could not be classified into the categories above. These numbers are resulting in a respective procedure-specific GIB-incidence of 1.1% for the LGIT, 7.7% for the UGIT and 8.9% for procedures in the HPB system during the study period. A diagram of the distribution of patients is shown in Figure 1. 

The localization of bleeding detected during emergency endoscopy was found in the UGIT in *n* = 46 (52.87%), in the LGIT in *n* = 18 (20.69%), and in the HPB system in *n* = 25 (28.73%) patients. In further detail, anastomotic bleeding that led to emergency endoscopy was found in *n* = 10 (11.49%). In relation to the associated surgical site, 22.22% of patients after UGIT procedures, 10.89% of patients after LGIT procedures, and 5.55% of patients after an HPB procedure were suffering from bleeding events in the region of their primary anastomosis. In all other patients, the bleeding event was not located in the area of the respective surgical procedures. Instead, gastroduodenal ulcerations occurred most frequently.

A detailed overview of the analyzed parameters is given in Table 4. Of the 108 patients undergoing emergency endoscopy, 80.55% (*n* = 87) had stigmata of gastrointestinal bleeding: 42 patients were found to have an active bleeding (38.89%), while 43 patients had evidence of bleeding that had occurred (39.81%). In addition, examinations of 21 further patients revealed no signs for a gastrointestinal bleeding event.

Anticoagulative therapy was documented in *n* = 63 patients (58.33%). Sources of GIB were gastroduodenal ulcerations (*n* = 45), bleeding esophagitis (*n* = 6), ischemic ulcerations (*n* = 13), hemorrhage after endoscopic sphincterotomy (*n* = 6), one case of variceal bleeding and one case with a bleeding gastric adenoma. Anastomotic bleeding was found in 9.26% of all analyzed patients. 

Emergency endoscopic therapy was successful in 83.8% of the cases. The most common endoscopic therapy in the patients studied was fibrin glue/suprarenin injection in combination with metal clips (*n* = 24; 44.44%). Injection monotherapy or clip monotherapy was performed in *n* = 10 patients (18.52%) each. The mean length of hospital stay for the total of 14 patients who underwent endoscopy before surgical intervention was 34.9 days. For the 94 other patients who underwent endoscopy subsequent to a surgical procedure, the mean length of hospital stay was 30.9 days. An endoscopically untreatable active bleeding situation at the time of emergency endoscopy existed in *n* = 5 of the patients who died in the further course. In cases of endoscopic untreatable bleeding situation, an angiographic intervention was performed in *n* = 4, or/and an additionally or secondary surgical procedure in *n* = 4 patients. 

The 30-day mortality was 9.26% (*n* = 10/108). All of these 10 patients were operated: *n* = 5 on UGIT, *n* = 2 on LGIT, and *n* = 1 on the HBP system. Two of the deceased patients could not be classified into the three respective surgical areas. The leading indication for surgery was mesenteric ischemia in *n* = 19. Of these patients, a number of *n* = 5 deceased in the clinical course. 

## 4. Discussion

The most common location of GIB events is the UGIT according to the literature. Here, performing emergency esophago-gastro-duodenoscopy (OGD) is recommended as the gold standard. According to Oakland et al., UGIB are more common than LGIB (33/100.00) in surgical patients with an incidence of 47/10,000 [1]. According to the study by Hebert et al., 2.3% of a total of 314 patients who underwent surgical procedures in the LGIT had postoperative bleeding events in the anastomotic area [6]. In contrast, our data shed new light on the incidence and location of GIB events in a surgical patient cohort. Since one would hypothesize to find the bleeding location in the surgical area of those patients, the majority of analyzed cases provided a different picture. Here, bleeding location outside the operated organ area was found in the majority of cases, while a classical anastomotic hemorrhage could only be detected in less than 10% of the cases (17% in the UGIT, 4% in HPB and 10% in the LGIT), which is the first critical finding of our study. In more detail, there was no marked difference between stapler anastomoses (UGIT and LGIT) and manual anastomoses (HPB), which is another interesting aspect of our findings.

The average age of surgical patients at the onset of GIB is reported to be around 67 years [7,8]. However, the cited patient cohorts were reported separately according to the procedures, for example, divided into patients with resections in the right or left colon or with bariatric upper abdominal procedures. Here, the latter ones are usually representing a younger patient population in contrast to cancer patients undergoing colonic resections. As a critical amendment to the cited literature, our retrospective analysis also included surgical patients who had undergone emergency endoscopy due to GIB already before surgery. Considering only patients who underwent endoscopy for GIB after surgery, the mean age in this subgroup was 63.9 years. Of note, when considering only patients who underwent emergency endoscopy and finally died during the clinical course, the mean age was 71 years. These results demonstrate the critical impact of age, thus providing another crucial aspect being helpful for the individual perioperative risk assessment of each individual patient.

The dichotomic classification of a surgical patient population into a pre- and a postoperative group was missing in the current literature yet. Here, most retrospective analyses of surgical patients were reporting the postoperative phase only [9,10]. In our study, however, patients with bleeding events prior to surgery were also included in the analysis in order to be able to indicate the number of bleeding-related surgical procedures despite primary endoscopic therapy. In this regard, more than half of the patients with bleeding-related endoscopy prior to surgery had to undergo surgery with the indication given by endoscopy. In contrast, in patients with postoperative bleeding-related emergency endoscopy, only one-third of the patients were suffering from acute GIB while one-third of patients displayed signs of past bleeding and the remaining patients had no bleeding stigmata. Of further clinical relevance, one-third of our surgical patients underwent an invasive procedure that finally provided no benefit to them, thus calling for critical clinical evaluation and indications for emergency endoscopy. 

Regarding the overall therapeutic success of endoscopic bleeding treatment, the current literature reports success rates in patient cohorts from internal medicine of approximately 80% or even higher. For example, the study by Jung et al. showed a success rate of acute endoscopic therapy of 88% [11]. Significantly lower success rates were reported by Pescatore et al. with 78.5% and 75.7%, respectively, when fibrin glue and epinephrine or epinephrine alone were used. [12]. In our study, endoscopic hemostasis could be achieved in 83.8% of cases, which is clearly within the range of the cited success rates for gastroenterological non-surgical patients. Of note, some GIB causes are not endoscopically reversible, for example in patients suffering from vascular ischemia representing a patient subgroup at highest risk with 50% mortality in our analysis. Nevertheless, these cases also represent “real-life” situations requiring first-line emergency endoscopy in surgical patients according to current emergency algorithms.

The 30-day mortality rate in the analyzed patient cohort is high and thus calling for sub-analysis to identify patients at highest risk. Here, half of the deceased patients were suffering from a mesenteric ischemia. The previously reported mortality rates in these patients is ranging from 60% up to 90% [13,14]. This high mortality rate shows the critical importance of established emergency algorithms including urgent endoscopic examinations also for surgical inpatients. In more detail, the high number of re-endoscopies is caused by relapse of bleeding, second-look endoscopies and unclear primary endoscopic results.

In summary, only a small number of surgical inhouse patients experienced a relevant GIB event. However, the associated 30-day mortality of GIB in the analyzed abdominal surgical patient population is increased at 7.75% when compared to the literature of gastroenterological patient cohorts [2,3]. Although GIB events in mostly heterogenous, postoperative patient cohorts have been poorly studied so far, the few data available report on a 30-day mortality are ranging between 0 and 13.3% for elective colonic/rectal resections [13,15]. Of note, classical anastomotic hemorrhage could only be detected in less than 10% of the cases irrespective of stapler or manual anastomosis, while for the majority of patients, the bleeding location was found to be outside the operated organ area. While our overall endoscopic therapy success rate was high and comparable to those achieved in non-surgical patients [11,12], especially vascular ischemia was not endoscopically reversible and linked to 50% specific mortality in our analysis [14,16].

The three key limitations of the present study are the retrospective and monocentric study design and the small number of cases, which are limiting the validity and generalizability of our results. Nevertheless, our results demonstrate that GIB events in surgical patients call for critical vigilance and require established, interdisciplinary emergency algorithms for rapid endoscopic diagnosis and therapy. Finally, a prospective, multicenter trial with a defined action plan in visceral surgery patients would be highly desirable.

## 5. Conclusions

Taken together, our study is the first to report the overall incidence of relevant gastrointestinal bleeding events in visceral surgical inpatients. Although the absolute number was rare, our analysis demonstrated several critical implications associated with the primary surgical area of the respective patients. Therefore, our data call for critical peri-operative vigilance for bleeding events and underscore the importance of interdisciplinary emergency algorithms.

## Figures and Tables

**Figure 1 jcm-12-01374-f001:**
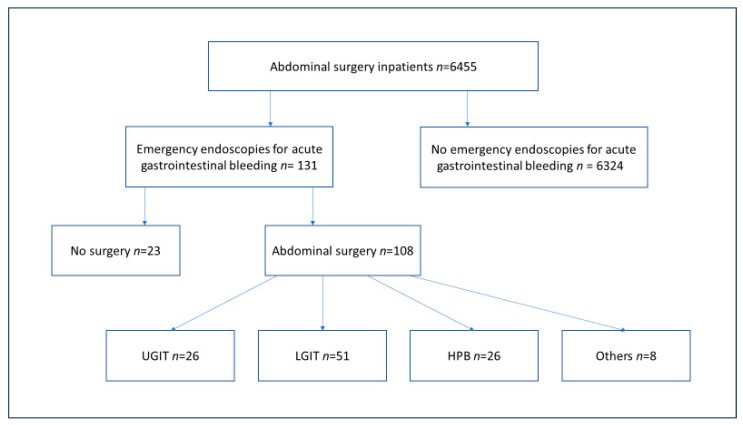
Patient’s distribution into the four groups regarding the operation area.

**Table 1 jcm-12-01374-t001:** Inclusion and exclusion criteria.

Inclusion Criteria	Exclusion Criteria
Bleeding relevant to circulation	No surgery
Hb loss greater than 2 g/dL	No emergency endoscopy
Need of transfusion	Emergency endoscopy without suspected bleeding
Need of interventions	Age < 18 years

**Table 2 jcm-12-01374-t002:** Classification of surgical interventions into four categories.

Operating Area	Included Surgical Procedures
Upper Gastrointestinal Tract (UGIT)	Esophageal, gastric or small bowel resections, bariatric surgery
Hepatobiliary System (HPB)	Liver, pancreas, bile duct resections, liver transplantation (LTx)
Lower Gastrointestinal Tract (LGIT)	Colonic, rectal resections, hemorrhoidal procedures.
Other	Hernias, PIPAC (Pressurized Intra Peritoneal Aerosol Chemotherapy)

**Table 3 jcm-12-01374-t003:** Patient’s characteristics.

Patients Groups Depended on Surgery	UGIT	HPB	LGIT	Other
Included patients (*n*)	26	23	51	8
Sex (m:f)	11:12	17:8	38:13	6:2
Age (years; $\overset{-}{X}$)	65.3	59.8	70.5	56.2
Surgery for malignancy (*n*; %)	14 (54)	10 (43)	25 (49)	0
Surgery for mesenteric ischemia (*n*; %)	8 (31)	1 (4)	5 (10)	2 (25)
Anticoagulation prior to surgery (*n*; %)	14 (54)	15 (65)	32 (63)	2 (25)

Abbreviations: UGIT = upper gastrointestinal surgery, HPB = hepatobiliary surgery, LGIT = lower gastrointestinal surgery, *n* = number, $\overset{-}{X}$ = average.

**Table 4 jcm-12-01374-t004:** Analyzed parameters of abdominal surgical patients with emergency endoscopies for bleeding.

Patients Groups Depended on Surgery	UGIT *n* = 23	HPB *n* = 26	LGIT *n* = 51	Others *n* = 8
Number of endoscopies per patient ($\overset{-}{X}$) (range)	2.62 (1–7)	2.40 (1–7)	1.64 (1–5)	1.38 (1–7)
Active bleeding (*n*) (%)	11 (48)	15 (58)	13 (26)	1 (13)
Signs of previous bleeding (*n*) (%)	4 (17)	4 (15)	16 (31)	2 (26)
Endoscopic no signs of bleeding (*n*) (%)	8 (35)	7 (27)	22 (43)	5 (65)
Anastomotic bleeding (*n*) (%)	4 (17)	1 (4)	5 (10)	0
Gastroduodenal ulceration (*n*) (%)	13 (57)	13 (50)	19 (37)	0
Ischemic ulceration (*n*) (%)	1 (4)	0	12 (24)	1 (13)
Hemorrhage after endoscopic sphincterotomy (*n*) (%)	1 (4)	1 (4)	3 (6)	0
Bleeding esophagitis (*n*) (%)	0	1 (4)	3 (6)	2 (25)
Variceal bleeding (*n*) (%)	1 (4)	0	0	0
Bleeding gastric adenoma (*n*) (%)	0	0	1 (2)	0
Success of endoscopic intervention (%)	67	84	90	94
Hemostatic procedure while urgent bleeding endoscopy				
(a) injection therapy	0	3	7	0
(b) clipping solely	3	3	4	0
(c) injection + clipping	10	7	6	1
(d) hemostatic powder	2	0	3	0
(e) stent	1	1	3	0
(f) variceal banding	1	0	0	0
(g) none	4	6	26	2
Endoscopy prior surgery (*n*) (%)	6 (20)	0	6 (12)	2 (25)
Length of hospital stay ( $\overset{-}{X}$, days)	31.00	39.16	31.25	4.31
30-day mortality (*n*) (%)	6 (26)	0	2 (4)	0

Abbreviations: UGIT = upper gastrointestinal surgery, HPB = hepatobiliary surgery, LGIT = lower gastrointestinal surgery, *n* = number,
$\overset{-}{X}$ = average.

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
