# Peer review of "Is There a High Risk for GI Bleeding Complications in Patients Undergoing Abdominal Surgery?"

_jcm, 2023, doi:10.3390/jcm12041374_

Round 1

Reviewer 1 Report

1. In the abstract, the indications for urgent enosdoscopy are incompletely described, and at the same time, the authors do not describe other exceptions, i.e. embolization and medical treatment.

2. The 30-day mortality is relatively high and should be evaluated and explained by the authors.

3. The authors do not explain which methods of stopping bleeding were performed during endoscopy, do not indicate the experience of the endoscopist and the conditions for emergency endoscopy. However, the authors do not describe the types of interventions, medical treatment of diseases that the patients had, which may affect the incidence of bleeding.

4. The article does not describe the type of surgical intervention in patients with mesenteric ischemia, which could have a significant impact on the interpretation of treatment results. Often, in addition to surgery, the incidence of bleeding can be affected by other factors (recurrent thrombosis, coagulopathies, etc.).

5. Endoscopic findings - analysis would be an important point of discussion, as based on the latter, we could conclude on the necessity of surgical revision and some other conditions.

6. In case of bleeding from anastomoses, the authors do not specify the type of anastomosis, used staplers/manual anastomoses, which would allow the present paper to have a useful and important discussion.

Author Response

  1. In the abstract, the indications for urgent endoscopy are incompletely described, and at the same time, the authors do not describe other exceptions, i.e. embolization and medical treatment.

# We would like to thank reviewer #1 for this helpful remark. We added the sentences:
“ Gastrointestinal bleeding (GIB) can cause life threatening situations. Here, endoscopy is the first line diagnostic and therapeutic mode in patients with GIB among further therapeutic approaches as embolization or medical treatment.”

  1. The 30-day mortality is relatively high and should be evaluated and explained by the authors.

# As mentioned by the reviewer, the 30-day mortality rate in our analysis is relatively high. Since the analyzed patient cohort of surgical inpatients has not been reported before, that is exactly the reason to present our data. Surgical patients are often in a reduced general condition and complications such as bleeding situations may lead to further clinical detoriation. To further explain this numbers, we inserted the following sentences in the discussion section: “The 30-day mortality rate in the analyzed patient cohort is high and thus calling for sub-analysis to identify patients at highest risk. Here, half of the deceased patients were suffering from a mesenteric ischemia. The previously reported mortality rates in these patients is ranging from 60% up to 90% [14, 15]. This high mortality rate shows the critical importance of established emergency algorithms including urgent endoscopic examinations also for surgical inpatients. In more detail, the high number of re-endoscopies is caused by relapse of bleeding, second-look endoscopies and unclear primary endoscopic results.”

  1. The authors do not explain which methods of stopping bleeding were performed during endoscopy, do not indicate the experience of the endoscopist and the conditions for emergency endoscopy. However, the authors do not describe the types of interventions, medical treatment of diseases that the patients had, which may affect the incidence of bleeding.

# Thank you very much for helpful advice. Please find patient’s information now in Table 3. Information on endoscopic findings are listed in Table 4.

  1. The article does not describe the type of surgical intervention in patients with mesenteric ischemia, which could have a significant impact on the interpretation of treatment results. Often, in addition to surgery, the incidence of bleeding can be affected by other factors (recurrent thrombosis, coagulopathies, etc.).

# We are grateful for this advice and have inserted this information now in Table 4. The following sentences are included in the results part: “The leading indication for surgery was mesenteric ischemia in n=19. Of these patients a number of n=5 deceased in the clinical course.”

  1. Endoscopic findings - analysis would be an important point of discussion, as based on the latter, we could conclude on the necessity of surgical revision and some other conditions.

# Thank you very much for this important comment. We have now included this information in Table 4 and changed the discussion section accordingly. 

  1. In case of bleeding from anastomoses, the authors do not specify the type of anastomosis, used staplers/manual anastomoses, which would allow the present paper to have a useful and important discussion.

# Thank you very much for this remark. We found no differences in the low number of anastomotic bleedings, but we clarified the bleeding sources, the endoscopic therapy and the therapeutic success rate. We revised the discussion section with focus on these findings.

Reviewer 2 Report

I would like to congratulate the authors on their research and the manuscript.

I feel that that certain things can be modified in the study design, result reporting and analysis which would make it as stronger manuscript. These are as follows:

1. Study design: the authors are looking at incidence and outcomes of significant GI bleeding in their cohort of surgical patients. I feel the authors should do some basic analysis with chi square tests and possible MVA to identify predictors of bleeding. Ideally, they should compare these to case matched non bleeding controls selected randomly.

2. Result reporting: while the authors have alluded to some results in the discussion section, they need to better report the following in the results section: Baseline demographics of patients, anticoagulant use etc. They should also report cause for GI bleeding and tabulate outcomes based on cause. They have differentiated anastomotic vs non-anastomotic bleeding but given very little detail about causes of non-anastomotic bleeding. Outcomes based on intervention are not reported. If most mortality occurred from vascular ischemia ( as against hmge), that is not a endoscopically reversible cause of GI bleeding and therefore should be highlighted as the prime cause and discussed in greater detail.

3. The discussion is broad, but needs to be focused so that the reader can have some concrete take home points.

4. It is possible that some of the points that the authors were trying to make have been not communicated appropriately and may have been lost in translation to English and therefore appropriate English language editing is also suggested.

Author Response

I would like to congratulate the authors on their research and the manuscript.

I feel that certain things can be modified in the study design, result reporting and analysis which would make it as stronger manuscript. These are as follows:

  1. Study design: the authors are looking at incidence and outcomes of significant GI bleeding in their cohort of surgical patients. I feel the authors should do some basic analysis with chi square tests and possible MVA to identify predictors of bleeding. Ideally, they should compare these to case matched non-bleeding controls selected randomly.

# We would like to thank reviewer #2 for this suggestion. However, this work is a pure retrospective and descriptive analysis being the first to address the GIB incidence in surgical inhouse patients. At this point, we do not have the required data to perform a statistical analysis with a matched pair analysis, which would be beyond the scope of this study but a great suggestion for further exploration and studies on this topic.

  1. Result reporting: while the authors have alluded to some results in the discussion section, they need to better report the following in the results section: Baseline demographics of patients, anticoagulant use etc.

# Thank you very much for this helpful advice. We have now included the mentioned data to the patient’s characteristics in Table 3.

They should also report cause for GI bleeding and tabulate outcomes based on cause. They have differentiated anastomotic vs non-anastomotic bleeding but given very little detail about causes of non-anastomotic bleeding. Outcomes based on intervention are not reported. If most mortality occurred from vascular ischemia (as against hmge), that is not an endoscopically reversible cause of GI bleeding and therefore should be highlighted as the prime cause and discussed in greater detail.

# Thank you very much for bringing up this point. We have included the following sentence and discuss this point in the discussion section in the following paragraph. Indeed, some GIB causes are not endoscopically reversible, for example in patients suffering from vascular ischemia. Nevertheless, this also represents “real-life” situations requiring first line emergency endoscopy in surgical patients according to current emergency algorithms.

  1. The discussion is broad, but needs to be focused so that the reader can have some concrete take home points.

# We adopted the discussion section with focus on the presented aspects.

  1. It is possible that some of the points that the authors were trying to make have been not communicated appropriately and may have been lost in translation to English and therefore appropriate English language editing is also suggested.

# Thank you very much for this advice. We thoroughly revised the language of our manuscript accordingly.

Reviewer 3 Report

Thank you for this article.

I think mixing both patients bleeding before surgery and after surgery introduces confusion as they are completely different entities with different bleeding etiologies.  it would be better to leave this cohort out as it does not add to the overall aim of the study.

it would also be helpful to list the findings beyond anastamotic bleeding.  it would be helpful to compare therapeutic success of the gastroenterologists for non-surgical bleeding over that same period of time.

in those who did not have successful endoscopic control, how many went to surgery or interventional radiology.

In the results section your first sentence is very confusing.  "During the study period, bleeding events with indication for emergency endoscopy occurred in 2.0% (129/6455) of all surgical inhouse patients. Of these 129 patients, a total of 83.7% (n=108) underwent surgery, while 21 patients (16.0%) underwent emergency endoscopy on the surgical ward without documented surgical procedures during the same inpatient stay".  It should be clear that the 108 patients had surgery before the endoscopic procedure.

Also, "A total of 14 (12, 96%) patients underwent endoscopy due to a GIB prior to surgery, where endoscopy was leading to the indication for surgery in more than 50% (8/ 14) of the 98 patients. For the vast majority of our inhouse patients (n=94; 87.01 %), emergency endoscopy due to suspected GIB took place after surgery."

what does the (12, 96%) reflect?

why do all goups have a high number of post-surgical endoscopies.  DO you have a range or a median?

You also discuss post-sphincterotomy bleeding.  Which types of surgical procedures were those found in?

Author Response

Thank you for this article.

I think mixing both patients bleeding before surgery and after surgery introduces confusion as they are completely different entities with different bleeding etiologies.  it would be better to leave this cohort out as it does not add to the overall aim of the study.

# Thank you very much for bringing up this point. When we designed this study, we were aiming to provide comprehensive data on emergency endoscopies for suspected GIB events in all patients on our surgical wards. We agree, that one could separate both entities (pre- and post-surgery), what we did in terms of subgroups analyses. However, we would not like to remove the pre-surgery group since we intended to present both subgroups reflecting a truly realistic picture of emergency cases other colleagues may encounter in daily life.  

It would also be helpful to list the findings beyond anastomotic bleeding. It would be helpful to compare therapeutic success of the gastroenterologists for non-surgical bleeding over that same period of time.

# Thank you very much for this tip. We included a description of endoscopic findings and the applied therapeutic procedure in Table 4. The therapeutic success of gastroenterologists for non-surgical bleeding is discussed according to the current literature. 

in those who did not have successful endoscopic control, how many went to surgery or interventional radiology.

# We would like to thank reviewer #2 for this interesting und helpful question. Here, we included the Table 3 with information on the therapeutic success and further treatment in patients without primary endoscopic hemostasis.

In the results section your first sentence is very confusing.  "During the study period, bleeding events with indication for emergency endoscopy occurred in 2.0% (129/6455) of all surgical inhouse patients. Of these 129 patients, a total of 83.7% (n=108) underwent surgery, while 21 patients (16.0%) underwent emergency endoscopy on the surgical ward without documented surgical procedures during the same inpatient stay".  It should be clear that the 108 patients had surgery before the endoscopic procedure.

# Thank you very much for bringing this point to our attention. We agree, that the previous sentence may lead to confusion. Thus, we clarified the numbers and included the following sentences into the results section: “During the study period, bleeding events with indication for emergency endoscopy occurred in 2.0% (129/6455) of all surgical inhouse patients. Of these 129 patients, a total of 83.7% (n=108) underwent surgery, while 21 patients (16.0%) underwent emergency endoscopy on the surgical ward without documented surgical procedures during the same inpatient stay. The patients without surgical procedure were excluded from further analyses. Of the analyzed 108 patients undergoing surgery and emergency endoscopy for suspected GIB events, n=94 (87.01%) were examined after surgery while n=14 (12.96%) patients underwent endoscopy prior to surgery.”

Also, "A total of 14 (12, 96%) patients underwent endoscopy due to a GIB prior to surgery, where endoscopy was leading to the indication for surgery in more than 50% (8/ 14) of the 98 patients. For the vast majority of our inhouse patients (n=94; 87.01 %), emergency endoscopy due to suspected GIB took place after surgery." what does the (12, 96%) reflect?

# Thank you very much for this question. These are 14 out of 108 patients with endoscopy for GIB prior to surgery, which comes to 12.96% of 108 patients in total. We added the total number in the first sentence to avoid confusion.

Why do all groups have a high number of post-surgical endoscopies. Do you have a range or a median?

# The range of the number of endoscopic examinations in the analyzed patients is 1 to 8 in the UGIT, 1-7 in the HPB, 1-5 in the LGIT, and 1-10 in the “others” group. This information is included in Table 3 now.

You also discuss post-sphincterotomy bleeding. Which types of surgical procedures were those found in?

# Thank you very much for this good question. We are happy to share this information with you:

  • In one patient with mesenteric ischemia a hemicolectomy with combined cholecystectomy was performed. In this case the patient developed cystic stump insufficiency and underwent ERCP. Finally, a metal stent was inserted for hemostasis.
  • In another patient with CED and secondary sclerosing cholangitis a proctocolectomy was performed. Because of increasing cholestatic parameters in the postoperative course an ERCP was performed with resulted post-sphincterotomy bleeding.
  • In the third patient a complex two-stage hepatectomy (ALLPS = Associating Liver Partition with Portal vein ligation for Staged hepatectomy) was performed. Because of increasing cholestasis in the postoperative course, the ERCP was performed.
  • The next patient underwent a sigma-resection for cancer and developed an acute cholangitis in the postoperative course.
  • In one patient a gastric wedge resection was performed for a neuroendocrine tumor. This patient developed an acute cholangitis in the postoperative course because of choledocholithiasis, not as complication of the surgery. He underwent an ERCP and developed a post-sphincterotomy bleeding.

We would also be happy to share this information with the readership, however we feel that this information would stretch the manuscript to much while focusing on a less than 5% on the included patients with >50% non-surgical related indications to ERCP.

Round 2

Reviewer 1 Report

-

Reviewer 3 Report

thank you for your edits.  i feel this is sufficiently revised.